# Assessing Extensive Semi-Arid Rangeland Beef Cow–Calf Welfare in Namibia: Part 1: Comparison between Farm Production System’s Effect on the Welfare of Beef Cows

**DOI:** 10.3390/ani11010165

**Published:** 2021-01-12

**Authors:** Yolande Baby Kaurivi, Richard Laven, Tim Parkinson, Rebecca Hickson, Kevin Stafford

**Affiliations:** 1School of Veterinary Medicine, Massey University, Private Bag 11 222, Palmerston North 4442, New Zealand; r.laven@massey.ac.nz (R.L.); t.j.parkinson@massey.ac.nz (T.P.); 2School of Agriculture and Environmental Management, Massey University, Private Bag 11 222, Palmerston North 4442, New Zealand; R.Hickson@massey.ac.nz (R.H.); k.j.stafford@massey.ac.nz (K.S.)

**Keywords:** animal welfare assessment, beef cow systems, semi-arid rangelands, Namibia

## Abstract

**Simple Summary:**

Namibia is in the process of updating animal welfare legislation. This needs to include an assessment protocol for beef cattle production systems that is sufficiently rigorous for the country to gain and maintain access to high-value beef export markets. Beef is produced in commercial and semi-commercial systems and in communal village farms. Privately owned commercial farms allow maximum herd and rangeland management to ensure optimum productivity and profitability. Village farms (semi-commercial and communal) have limited grazing land, with consequent challenges of grazing and water management, as well as traditional customs of cattle management. A protocol was developed to assess the welfare of beef cattle in the context of these production systems. The application of the welfare assessment protocol indicated that the standards of welfare differed across production systems, with commercial farms achieving the best standard of welfare, followed by semi-commercial, then communal village farms. The greatest opportunity for change exists within the semi-commercial village farms, which need to attain to the requirements imposed by international markets to maximize their returns; hence herd management and welfare status is better than in the purely communal farms. This suggests that commercialization of communal farming may have benefits for animal welfare.

**Abstract:**

A proposed animal welfare assessment protocol for semi-arid rangeland-based cow–calf systems in Namibia combined 40 measures from a protocol developed for beef cattle in New Zealand with additional Namibia-specific measures. Preliminary validation of the protocol had been undertaken with five herds in one semi-commercial village. The aim of the current study was to apply this protocol and compare animal welfare across three cow–calf production systems in Namibia. A total of 2529 beef cows were evaluated during pregnancy testing in the yards of 17 commercial, 20 semi-commercial, and 18 communal (total: 55) herds followed by an assessment of farm resources and a questionnaire-guided interview. Non-parametric tests were used to evaluate the difference in the welfare scores between the production systems. The results indicated a discrepancy of animal welfare between the three farm types, with a marked separation of commercial farms from semi-commercial, and communal village farms in the least. The differences in these production systems were mainly driven by economic gains through access to better beef export market for commercial farms and semi-commercial villages, as well as by the differences in the available grazing land, facility designs/quality, and traditional customs in the village systems. The results indicate an advantage of commercialization over communalization.

## 1. Introduction

Namibian beef farming is dependent on extensive grazing under semi-arid, rain-fed conditions [1,2]. The beef cattle industry is predominantly driven by the demarcation of Namibia into two parts (northern and southern) by the veterinary cordon fence (VCF; see Figure 1). The VCF is an important biosecurity measure for animal disease control, particularly for contagious bovine pleuropneumonia and foot and mouth disease (FMD). North of the VCF, livestock and livestock products are not eligible for overseas export because of the risk of FMD which is endemic in north-east Namibia [3,4]. All farming rangeland north of the VCF is allocated to communal farmers with customary tenure [5,6]. Most households in these areas are subsistence-based and labor intensive, with inadequate access to technology [6]. 

The southern part of Namibia (below the VCF) was declared FMD free by the World Organisation for Animal Health (OIE) and can export beef to overseas markets [4]. In this zone there are two types of farms; privately owned commercial farms [2] and communally owned village farms with customary tenure. The latter “semi-commercial” village farms are set-up in the same way as communal village farms north of the VCF but have access to high-value commercial and overseas markets [6,7]. The line of these two systems is also blurred by the resettlement freehold farms where the Namibian government buys farms from commercial farmers and resettle people mostly from communal areas.

These three different beef farming systems, commercial, semi-commercial and communal, have different management styles, income streams, and levels of productivity. Commercial farms are farms that are not communally owned, have livestock grazing behind fences, and where the focus of beef farming is principally commercial rather than being a traditional tribal practice. In contrast, both semi-commercial and communal villagers farm in communal areas. The main difference between the two is that semi-commercial farms have access to commercial markets and thus have different cattle marketing and management strategies, but in both systems, traditional tribal practices in relation to cattle farming are important. In this paper we use the term village farms as an inclusive term for both semi-commercial and communal farms, as a way of contrasting these communal farms to commercial beef farms.

These differences are likely to have a significant effect on cattle welfare, although there are no data on beef cattle welfare on Namibian beef farms. In part, this dearth of information is because there is no standardized system of independent assessment of animal welfare under Namibian production systems. However, welfare assessment is commonly demanded by the type of markets to which Namibian farm animal products are exported. Importing countries commonly have welfare assessments that are applied to their domestic beef production, but such assessments are focused upon the species and production systems that pertain in those countries [8,9,10], and do not take into consideration the conditions that pertain in (for example) the semi-arid conditions of countries such as Namibia. Thus, as different systems need different welfare assessment protocols [11,12], the value of developing a system of welfare assessment that was relevant to Namibian conditions became evident.

As part of a project to create welfare assessment protocols for extensively reared beef cattle in the semi-arid conditions of Namibia, Kaurivi [13] assessed the use of a protocol that was originally developed for New Zealand semi-intensive temperate pastoral beef production systems (based on the Welfare Quality protocol [14] and University of California Davis protocol [15]). The protocol was validated on five herds in one semi-commercial village in Namibia. That preliminary study identified nine further Namibian-specific measures that could be incorporated into the protocol to make it suitable for use on Namibian beef farms. By using the protocol on commercial, semi-commercial, and communal beef farms in Namibia, the aim of this study, was to confirm that conclusion and to determine the impact of farm system on the welfare of Namibian beef cows.

## 2. Materials and Methods

Three areas were selected for inclusion in the study based on being representative of their farm type within Namibia. The herds included in the study were a convenience selection based on the willingness of the farmers to be involved. A total of 55 herds were enrolled, 17 commercial herds (from 17 separate farms), 20 semi-commercial herds (from 8 villages) and 18 communal herds (from 8 villages). Thus, both semi-commercial and communal farms were village-based, with multiple herds examined per village. Details of these herds are given in Table 1.

### 2.1. Description of the Study Areas

Commercial farming system: (Gobabis area, Omaheke region, eastern Namibia, lat 22°26′ S, long 18°58′ E). Commercial farms are large farms (range 3000–10,000 ha) that have a boundary fence and fenced grazing paddocks [16]. On most farms (every farm represents one herd), the owners are full-time farmers or employ managers [2]. On the study farms, beef cattle farming was the predominant economic activity of interest, through auctioning or direct abattoir marketing of cattle. Cattle farming was supplemented by small stock (sheep and goats) farming and wildlife ranching. Horses were commonly used for cattle mustering on most farms. The principal beef breed was Brahman, alongside pure breeds or crosses of Simmental, Charolaise, Bonsmara, Afrikaner, and Nguni. The area is a flat sandy thorn-bush highland with predominant open acacia savannah-type vegetation [17]. Annual rainfall in the area ranged between 200–400 mm (~2/3 in January–March) and average annual temperature is 19.3 °C; ranging from 0 °C in winter to 33 °C in summer [18]. 

Semi-commercial farming system: (Okakarara area, Otjozondjupa region, central Namibia, lat 20°35′ S, long 17°27′ E). In a semi-commercial village farm, multiple families use the same permanent communal land for grazing and water. The land is government-owned with limited or no internal and external border fencing and animals roam freely [5,19]. Semi-commercial village farmers in Okakarara are mainly dependent on livestock production (multi-purpose cattle, sheep and goats) for their livelihood. Weaner calf trading at auctions was the main marketing activity. In the study area, the grazing area associated with an individual village ranged from ~3000 to 6000 ha, with ~10–30 families/households per village. Herds were selected for inclusion on a convenience basis with 1–5 herds selected from each of eight separate villages (total *n* = 20 herds; Table 1). In one of the eight study villages, government-built, village-maintained, common cattle handling facilities (yard with a race) were available; in the other villages, farmers were responsible for erecting their own facilities. Some households, without adequate handling facilities, took their cattle to yards with better facilities on days of mass handling. The principal beef breed was Brahman, alongside crosses with Simmental, Hereford, Nguni, and Sanga. The area is a sandy thorn-bush highland acacia savannah [7]. Annual rainfall in the area ranged between 100–430 mm (~2/3 in January–March) and average annual temperature was 19.6 °C; ranging from 0 °C in winter to 32 °C in summer [18]. 

Communal farming system: (Opuwo/Kaokoland area, Kunene region, north-western Namibia, lat 18°3′ S, long 13°51′ E). As in semi-commercial villages, in communal villages, animals of different households graze together on the same government-owned communal land with limited or no internal and external border fencing. The small cattle herds were mainly kept for subsistence purposes and were mostly multi-purpose; i.e., milk and meat as well as manure for fire and building houses. Cattle farming was supplemented by herds of goats and some sheep. In the study area, farmers tend to be semi-nomadic with livestock being moved from permanent structures (yards and water troughs) to temporary ones in seasons of limited grazing (e.g., winter or drought). Cattle marketing was mainly for oxen (more than 2 years old bullocks) and cattle must go through a quarantining system, before slaughter at an approved abattoir a long distance away. In this area, the government had constructed community crush pens (forcing pen and race only) within ~3 km of each village. These were completed and maintained by the villagers. The main breed in the area was indigenous Sanga cattle alongside Nguni and Brahman crosses. Average cattle herd size was ~ 254 cattle, ranging from 30–548 cattle (one herd with 1900 cattle was owned by multiple members of the same family but managed as one group). From the selected eight villages, a selection of 2–5 herds per village farm was included in the study (total *n* = 18 herds; Table 1). The area vegetation is classified as mopane savannah with shrubland-woodland mosaic vegetation partly with rocky and bare mountains [20]. Annual rainfall in the area ranged between 50–320 mm (~2/3 in January–March) and average annual temperature was 21.6 °C; ranging from 10 °C in winter to 30 °C in summer [18].

### 2.2. Welfare Assessment and Data Collection

The welfare assessments took place in March/April 2019 (autumn). The protocol was used on 55 herds with animal-based measures being assessed on 2529 cows (Table 1). (See Appendix A for protocol and description of measures). All animal-based and handling assessments were made during pregnancy testing, with all cows presented for pregnancy testing being assessed on each herd. On farms where a race was present, pregnancy testing was undertaken in the race. In the absence of a race, cows were captured with ropes (as if for milking) and pregnancy tested while standing (or lying) in a holding pen. 

All observations were made by the same observer (first author). For each herd, the observer took a general overview of the cows in holding pens, before observations of body condition, rumen fill, behavior, and physical health were made in single-file races (or in pens where there were no races). Stockpersonship was evaluated as cows entered, were handled and exited the race or the pens. Information was collected on yard design and accessibility (i.e., shape and size of forcing pens, race structure as well as cow flow and effective handling from pens to race). As animals exited the race (or pens), their exit speed (running or walking), whether they fell or stumbled, and lameness signs were all recorded. Depending on accessibility and race design, the position of the assessor varied from standing on the side of the race or in pens as cattle moved around. For observation of cows exiting the race/pens the observer stood as close to the exit as possible without interfering with cow flow.

A farm resource visit and a questionnaire-guided interview were conducted to assess health and management of each herd over the last 12 months. These included records of dehorning/disbudding, castration, vaccination, diseases, or disease symptoms seen in cattle, cattle deaths, access to water and grazing and wintering practices. (See Appendix B for questionnaire).

### 2.3. Data Analysis

All data were analyzed using IBM SPSS Statistics for Windows Version 24 (IBM Corp. Released 2016. Armonk, New York, USA). Descriptive statistics for continuous measures were used to capture central tendency (median), and range (minimum and maximum). The effect of farm type on continuous measures of welfare was analyzed by using the Kruskal-Wallis test, and on categorical measures using the Fisher’s exact test. Where the *p*-value was <0.2 for either test, post-hoc testing was used for pairwise comparisons (Dunn test for continuous measures, and Fisher’s exact for categorical). Holm-Bonferroni correction was used to account for multiple comparisons. For this analysis α was set at 0.05.

## 3. Results

The proposed protocol took on average 2.5 h for a 100-cow herd at commercial farms (yard assessment-1 hour and questionnaire and farm resource visit-1.5 h) and 2.5 h for a 50-cow herd at the village farms (semi-commercial + communal).

### 3.1. Continuous Measures

The median and range for the 30 continuous indicators included in the assessment are shown in Table 2 (mean ranks for these measures are shown in Appendix C). Of those measures, only one, broken tails, had no recording on any farm of any type. Five measures (dirtiness, blindness, ocular and nasal discharge and poisoning deaths) had medians of 0 on all three farm types. Of the other 24 measures, commercial herds had the lowest median for 15 (including one tie) and the highest median for seven (including one tie). Semi-commercial herds had the lowest median for 9 (including three ties) and highest median for two. In contrast communal herds had the lowest median for three measures (including two ties) and highest median for 16 measures (including one tie).

No effect of farm type was identified on the numbers of cattle affected with 14/24 measures (swelling, blindness, ocular and nasal discharges, dystocia, fly burden, deaths from disease, accident, culling or poisoning, fearful behavior, falling/lying, stumbling or running (*p* ≥ 0.35)). Of the measures where the Kruskal-Wallis test returned a *p*-value of 0.05 to 0.2 (i.e., blindness, fly burden, run and excess branding/wounds), none could be separated by farm category using the Dunn test. As such, the lowest adjusted *p*-value for a pairwise comparison was >0.0167 in all cases, as there were 3 pairwise comparisons (commercial vs. semi-commercial and communal and semi-commercial vs. communal), and using the Holm-Bonferroni correction all pairwise comparisons are statistically non-significant if the lowest *p*-value is > 0.05/3.

For the remaining measures, grouping by farm category using the Dunn test is shown in Table 3. The analysis separated the three farm types for proportion of emaciated cows and proportion of cows with poor rumen fill. In both cases, the median was lowest on commercial herds (0%) and highest in communal herds (83.9% and 78.9%), with semi-commercial herds (13% and 48.1%) in between the two. For long/sharp horns, hair loss, tick burden and mortality, commercial herds were separated by the analysis from semi-commercial and communal herds, but the analysis did not separate the village farms. In all cases, the median was lowest on commercial herds (see Table 2). 

The analysis for dirtiness showed no separation between commercial and semi-commercial herds or between commercial and communal herds but separated semi-commercial from communal herds. Although in all farm types, the median was 0, semi-commercial herds had a lower mean rank than communal herds. The comparison of farm effects for diarrhea and reproduction deaths separated semi-commercial from commercial and communal herds but did not separate commercial from communal herds. In both cases, semi-commercial had the lowest median value and communal had the highest (Table 2). 

There was an effect of farm type on the proportion of cows with skin abrasion and lameness, and the number of reported predator deaths. The analysis separated commercial and semi-commercial herds from communal herds, but not commercial from semi-commercial herds. In both these cases the median was highest in communal herds (Table 2). For deaths associated with nutritional deficiency, the analysis separated semi-commercial herds (which had the highest median) from commercial and communal herds but did not separate communal and commercial herds.

### 3.2. Categorical Measures

Shade (natural savannah-type rangeland trees) was sufficient on all the farm types. The remaining frequencies of ordinal measures by farm type in each welfare category are shown in Appendix D. Commercial herds had the highest frequency of farms in the poor welfare category for 4/17 measures (dog noise, equipment noise, mis-catch, and use of electric prodders). Semi-commercial village herds had the highest frequency of farms in the poor welfare category for 4/17 measures (late castration and dehorning, hazards, and poor flow of cattle during handling). Communal herds had the highest frequency of farms in the poor welfare category for 8/17 measures (distance to water/grazing, ear tagging, hitting and tail twisting in the yards, handler noise, and yarding frequency).

Of the categorical measures, no effect of farm type was found on 4/17 measures (hazards, hitting, equipment noise, and health checks; *p* >0.2 in the initial analysis). The *p*-values for the multiple comparisons and direction of the difference for the remaining measures are shown in Table 4. Although the overall *p*-value was <0.2 for mis-catch and dog noise, the analysis did not separate these measures by farm (lowest adjusted *p*-value >0.0167).

Electrical prodders were only used on commercial farms, so the analysis separated commercial herds from communal and semi-commercial herds, but not communal from semi-commercial. For distance to water/grazing, dehorning, castration, ear tagging, tail twisting and yard/handling flow, the analysis separated commercial from semi-commercial and communal herds, but not semi-commercial from communal herds. For all these measures, commercial herds had the lowest frequency of herds with poor welfare and communal the highest frequency.

For hot-iron branding, semi-commercial herds had the highest frequency of poor welfare. The analysis separated them from commercial and communal herds but did not separate communal from commercial herds. For noise of handlers, the analysis separated commercial and communal herds (with the latter having a higher proportion of farms with poor welfare than the former) but did not separate semi-commercial herds from communal or commercial herds. For yarding/handling frequency the analysis separated communal from commercial and semi-commercial herds, but not commercial from semi-commercial herds. For this measure communal herds had the highest of herds with poor welfare

Painful management procedures: Castration was performed on 9/17 commercial herds (mode and median 2 months of age; range 1 week to 8 months), 8/20 semi-commercial herds (mode and mean 6 months) and on all 18 communal herds (all after 6 months). Disbudding was performed at all commercial herds (mode 2 months; 1 week to 8 months), 17/20 semi-commercial and 12/18 communal herds (mode 6 months at the villages; range 4 months to 12 months). Ear tagging was performed on all herds with median and mode of 2 months at the commercial herds and mode and median of 6 months in both types of village herds. Ear notching (cutting with a knife) was routinely performed on all village herds as a form of identification but not on the commercial herds. Skin cutting (dew-lap cuts) was performed on 7/20 and 4/18 herds in the semi-commercial and communal herds, respectively. Cattle branding with hot iron was used on all herds; secondary branding (i.e., letters, marks) was routinely performed on commercial stud farms (5/17 commercial farms) and on all the village herds. No anesthesia was used for any procedure.

## 4. Discussion

The current study confirmed the findings of Kaurivi [13] that a 40 measure protocol developed for use on extensive cow–calf farms in New Zealand was, with the addition of nine Namibian-specific measures, feasible for use during yarding on all classes of Namibian beef farms. The use of this protocol has identified marked differences between production systems in animal welfare. For 25 of the total 47 indicators (30 continuous + 17 categorical) included in the protocol, the analysis separated welfare outcome by system. For 16/25 measures, the welfare outcome was better on commercial herds than semi-commercial herds, and for 19/25 it was better on commercial herds than communal herds (Table 3 and Table 4). For 9/25 measures, semi-commercial herds had a better welfare outcome than communal herds, while communal herds had better outcomes overall than semi-commercial herds for only two measures. 

The most obvious difference between farm systems was related to feeding. For proportion of emaciated cattle and proportion with poor rumen fill, commercial farms performed better than the village farms, and semi-commercial farms performed better than communal farms. This is likely to be because commercial farms have more control over cattle feed supply than village farms. Village systems also have limited fencing, which limits the ability of farmers to manage grazing [1,19,21]. Moreover, social obligations and customary factors contribute to a widespread failure of village farmers to undertake timely reductions of stock numbers, even when feed is scarce [7]. Nevertheless, despite these challenges, semi-commercial farmers were able to maintain cattle in better body condition score (BCS) than their communal counterparts. This may be due to the better feeding, herd management, and genetics associated with the cash income from beef sales. On the other hand, it may reflect differences between the selected regions rather than farm system. Siegmund-Schultze [7] reported that the average BCS was higher on communal than semi-commercial village farms (3.0 vs. 2.5 respectively; 1–5 scale), concluding that this was related to feed supply, as the communal farms in northeast Namibia had significantly more rainfall than the semi-commercial farms in central Namibia (550 vs. 350 mm/year, respectively). Similar considerations may pertain to the present study, the communal farms were in northwest Namibia (Opuwo area), which has a much drier environment than the region (Kavango West) studied by Siegmund-Schultze [7]. Indeed, in the rainy season prior to the start of the study (i.e., September 2018–March 2019), Okakarara semi-commercial area had >140 mm more rain than Opuwo (457 vs. 314 mm, respectively). Thus, the differences of BCS in the present study may, at least partly, reflect the effect of rainfall on feed availability rather than system differences.

The distances that animals had to walk to grazing may also have exacerbated pressures upon BCS [22]. Distances are characteristically shorter on commercial farms (2–4 km) than on semi-commercial (3–6 km) or communal farms (4–8 km). The distance to grazing also had an impact on distance to water. In village farms, water is provided in or around the yards, so distance to water is strongly related to distance to grazing, whereas in commercial farms, water troughs are in the grazing paddocks. The relationship between BCS and lameness may therefore be of interest, since lameness was rare on commercial and semi-commercial herds (median 0% on both max. 3.9% and 3.3%, respectively), but more common on communal herds (median 3.3, max 17%). The association between the prevalence of lameness and distance walked is well recognized [23,24] and cattle on communal village farms walked the longest distances. Lameness rates could be also related to handling facilities, which could cause injury and acute lameness. However, there was no difference between semi-commercial and communal herds in yard handling/flow (Table 4), despite a difference in the prevalence of lameness prevalence. The older age of cattle at communal farms could also have contributed to lameness. Finally, BCS itself can affect the risk of lameness, as poorer BCS has been associated with increased risk of lameness [25], hence, the increased lameness in communal herds could have been related to underfeeding. There is clearly a need for further research to unravel these relationships.

Differences between systems were also noted in relation to animal health. Skin abrasions/injuries were more common in communal than semi-commercial and commercial herds, probably because the quality of handling facilities on communal farms were poorer (i.e., greater use of thorny bushes and tree poles tight with wires), and thus the risk of accidental injury higher. Hairless patches were less common in commercial than semi-commercial/communal farms, although this may reflect the prevalence of lumpy skin disease than of welfare *per se.* Relatively few animals were assessed as having diarrhea, although the prevalence was highest in communal than other herds (4.3% vs. 0% and 0.9% at semi-commercial and commercial respectively).

There was a large difference between the median of mortality rates on commercial (2.4%) versus semi-commercial (11.3%) and communal (11.7%) herds. It was likely that underlying differences between systems were exacerbated by the impact of the prevailing drought. Deaths due to predation, dystocia (reproduction) and nutrition varied between farm systems. Interestingly, death due to predation was higher in communal herds but there was no difference between commercial and semi-commercial herds. This may reflect the value of internal fencing and the ability of commercial farmers to control where cattle go (such as calving in calving camps in areas not frequented by cheetahs; [26]). Risk of death due to predation may also reflect changes in the population of wild prey associated with drought conditions [27,28]. Opuwo communal area borders the Etosha National Park where livestock are at a risk from predators. Deaths due to dystocia (reproduction deaths) were lowest in semi-commercial herds, but there was no difference between commercial and communal herds. However, the risk of deaths due to dystocia was not significantly related to the reported incidence of dystocia per se. There is no clear explanation for this, except to note that the rate of dystocia and reproduction deaths in communal herds was probably related to the lower plane of BCS [29]. It is likely that the higher rate of reproduction deaths on commercial farms was due to better recording of the cause of death as on commercial farms pregnant cows were kept in paddocks close to the main yards to allow for close monitoring and assistance in case of cows with difficult births.

Deaths reported as nutritional were far higher in semi-commercial than communal or commercial herds. Again, the reasons for this pattern are not clear, although the lower prevalence in commercial herds is consistent with better body condition and the use of supplementary feeding. Conversely, the lower prevalence on communal herds is not consistent with the higher proportion of emaciated cattle in those herds. It may be that the discrepancy lies in the effects of phosphate deficiency [30], which is widespread, but whose distribution and severity varies between the different areas included in the present study. For example, sandy soil around Okakarara areas has significantly lower phosphate than that at the harder and drier sand around Opuwo area (3.35 mg P/kg; [17], vs. 12.4 ± 1.7 mg P/kg; [20]), while the available soil phosphorus threshold is intermediate (≤ 10 mg kg -1 DM; [30]). A comprehensive investigation in the suspected phosphate deficiency related deaths of cattle is warranted and would help to unravel this question.

No effect of farm system on fly burden was found, but tick burdens were lowest on commercial farms. It is likely that this reflects parasite management with commercial farms using external parasite control as a key part of herd health management, whereas semi-commercial farmers treat cattle for ticks when necessary to do so (i.e., when preparing them for sale, when there are high tick burdens or in response to tick-borne diseases). No difference was found between semi-commercial and communal herds in tick burden, despite communal farmers rarely, if ever, using tick control. These data suggest that the limited use of tick control in semi-commercial herds is not having any impact on ticks.

There are issues with painful management procedures on all farm types. No anesthesia was used for any procedure on any farm. On commercial farms, dehorning/castration is generally performed when calves are younger than on village farm types (mode 2 vs. 6 months respectively). The FAN (Farm Assured Namibian) Meat scheme [31] requires that these operations are performed on calves younger than 2 months of age to ensure reduced pain and faster healing [32]. However, the FAN Meat is a voluntary scheme, and farmers in the villages have a traditional but erroneous conviction that late castration allows faster growth and muscle gain [33].

Keeping of horned cattle is traditional in the villages, but there is clear evidence of long horns injuring other animals and hampering handling flow in the race. There is increasingly strong advocacy for keeping polled cattle (which would also eliminate the need for dehorning: [34]). However, horns are natural defense tools against predators’ attacks [35] and imposing such restrictions in areas that lose livestock to predators may hinder the sustainability of farming.

Hot-iron branding and ear tagging are compulsory for cattle identification and traceability in Namibia. Analgesia was not provided on any farm for cattle during or after hot-iron branding, so hot iron branding was a significant welfare concern on all farms. However, on many farms additional painful methods were used to permanently identify cattle in addition to the statutory brand. On commercial farms, stud breeders (5/17) used an extra brand on stud cattle, while on village farms additional brands (e.g., letters, names, certain signs) as well as markings (e.g., dew-lap skin-flap cutting; see Appendix E) were used for security and easy identification. Rebranding of cattle that change ownership was also common. Branding was also used on village farms for treatment of musculoskeletal problems (e.g., hip tendon slipping), with cattle being branded over the affected site. Branding for security reasons was most common in semi-commercial areas, due to the high levels of stock theft in such areas. Furthermore, on village farms, ear notching (with knives), was used in addition to compulsory ear tagging because of the perception that ear tags were easily lost or dislodged. Thus, cattle identification, for whatever reason, was a welfare issue on all farms, especially village farms. On the other hand, the national traceability system (NamLITS) benefits cattle movement control [36], which is itself a critical underpinning of the Namibian beef industry. There would be great value in reducing the need to use painful methods for such identification.

There were differences between systems in measures related to stock handling. Only 6% of commercial herds had poor cattle flow/handling, compared to 50% and 70% in communal and semi-commercial herds, respectively. The effects of farm type on cattle handling were determined by differences in yard design and quality. For commercial farms with marginal flow, the main issues were related to yard design (e.g., oversized holding pens, sharp corners, and poor accessibility of the forcing pen). The problems on village farms were more fundamental, notably lack of gates, races, and forcing pens. In semi-commercial village farms yards were well constructed (usually) with wooden poles, whereas some communal villages yards were partially, or fully, constructed using thorny bushes. Thus, on village farms, poor construction and poor design of yards resulted in poor cow flow and subsequently increased handling times, frequency of hitting and tail twisting, and noise of handlers.

Despite differences across farm types in facilities and stock handling behavior, there was no farm type effect on behavior related to handling (fearful/agitated, fall/lie, stumble and run). This may be related to the “flighty” temperament of the Brahman breed which was predominant across all farm types [37]. For example, Brahman cattle and their crosses lay down in the race more commonly than other breeds, contributing to the high proportion of fall/lie cows in this study across all farm types.

Importantly, however, not all issues related to cow handling were worse on village farms. Electric prodders were only present on commercial farms (5/17); and all these farmers were using prodders on more than the 1% recommended by Grandin [38]. The use of prodders was related to cow flow in the yards with marginal designs and cattle lying in the race. The data from this study on stock handling suggest that improvements are needed across all farm types. In particular, training on alternatives to sticks, whips, pipes, and prods, such as flags on sticks [39,40] is needed. It also reinforced the value of a race in reducing stress during handling [41,42], especially for extensively reared cattle that are not frequently restrained [43].

## 5. Conclusions

The study evaluated the welfare of beef cattle in the various extensive beef production systems of Namibia. It showed that the welfare of cows varied between farm types: 25 of the 44 criteria in the present study varied across systems. Of those 25 measures, the welfare outcome was better on commercial herds than semi-commercial herds in 16 and better than communal in 19. The main reason for the better performance of commercial herds were better nutrition and management of grazing land, better cattle handling infrastructure, and preventative disease control. Despite the similar animal welfare challenges facing semi-commercial and their communal counterparts, of the 25 measures found to be affected by farm type, 9 were better on semi-commercial herds than communal herds and only 2 were better on communal than semi-commercial herds. This suggests that even limited commercialization of communal farming may have benefits for animal welfare. However, this conclusion needs testing on more farms in more areas of Namibia. There is an immediate need however, to categorize measures across the farm production systems to indicate thresholds of acceptable and unacceptable welfare, to give guidance regarding the levels at which intervention and remediation is required. This will be addressed in a companion Part 2 paper of the current study.

## Figures and Tables

**Figure 1 animals-11-00165-f001:**
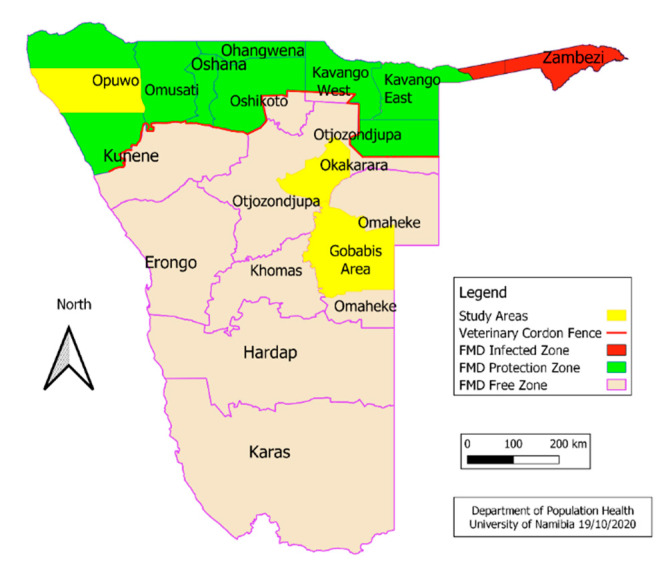
Namibia regional map showing the foot and mouth disease (FMD) zones and fences and study areas for commercial (Gobabis area), semi-commercial (Okakarara area) and communal farming (Opuwo area) [4], (source Dr Alec Bishi).

**Table 1 animals-11-00165-t001:** Study areas for commercial, semi-commercial, and communal farming areas showing number of herds, farms/villages and number of cows assessed.

Beef Cattle System	Area	No. of Herds Assessed	No. of Farms/Villages	Total Cattle	Average Cattle/Herd	Total Cow/Heifers	No of Cows/Heifers Assessed			
							Average	Min	Max	Total
Commercial	Gobabis	17	17	5887	346	3923	76.7	34	141	1305
Semi-commercial	Okakarara	20	8	2196	110	1590	29.7	11	55	593
Communal	Opuwo	18	8	4563	254	2485	35.1	8	78	631

Both semi-commercial and communal villagers farm in communal areas (village farms). The main difference between the two is that semi-commercial farms have access to commercial markets and thus have different cattle marketing and management strategies. In this paper, we use the term “village farms” as an inclusive term to include both semi-commercial and communal farms, to contrast them with commercial farms which are not communally owned, which have livestock grazing behind fences, and where the focus of beef farming is principally commercial rather than being a traditional tribal practice.

**Table 2 animals-11-00165-t002:** Descriptive analysis (median and range in percentage) of continuous welfare indicators in three Namibian beef production systems.

Measure	Commercial (*n* = 17)	Semi-Commercial (*n* = 20)	Communal (*n* = 18)
Thin cows	2.5 (0–46.9)	78.3 (25–100)	**100** (83.3–100)
Emaciated cows	0	7.3 (0–40)	**83.9** (52.5–100)
Poor rumen fill	0 (0–45.2)	48.1 (14.3–100)	**78.9** (39.3–100)
Dirtiness	0 (0–8.8)	0	0 (0–16.7)
Swelling	1.7 (0–8.2)	**3.7** (0–37.1)	2.3 (0–13.3)
Hair loss	0 (0–2.8)	2.8 (0–16.7)	**5.3** (0–20)
Abrasion	2.8 (0–12.5)	5.6 (0–27.3)	**19** (1.8–40)
Multiple brands/wounds/cuts	0.8 (0–3.6)	2.8 (0–44.4)	**5.3** (0–87.5)
Broken tail	0	0	0
Long/sharp horns	2.5 (0–37)	40.5 (10–96.6)	**61.8** (13.1–85)
Blindness	0 (0–2.5)	0	0 (0–1.3)
Ocular discharge	0 (0–1.4)	0	0 (0–1.3)
Nasal discharge	0 (0–1.4)	0	0 (0–1.3)
Diarrhea	0.9 (0–12.9)	0 (0–1.8)	**4.3** (0–25)
Lameness	0 (0–3.9)	0 (0–3.3)	**3.4** (0–16.7)
Dystocia	0.8 (0–3.3)	1 (0–10.0)	**1.4** (0–6.7)
Tick burden	0 (0–6.8)	1.4 (0–81.3)	**4.4** (0–35)
Fly burden	0 (0–12.3)	1.4 (0–61.5)	**4.5** (0–55)
Deaths from diseases	**0.7** (0–4.1)	0 (0–2)	0.4 (0–7.9)
Accidental deaths	**0.2** (0–1.6)	0 (0–3)	0.1 (0–16)
Culling for health	**0.2** (0–2.3)	0 (0–3.1)	0 (0–3.3)
Predation/snake bite deaths	0.4 (0–9)	0 (0–4.1)	**4** (0–20)
*Nutritional deaths	0 (0–3.2)	**6.6** (0–50)	1.1 (0–10)
*Poisoning deaths	0	0 (0–2.4)	0 (0–0.3)
*Reproduction deaths	**0.2** (0–0.8)	0 (0–4)	**0.2** (0–2.6)
Annual mortality rate	2.4 (0–15.5)	11.3 (0–26.9)	**11.7** (2.2–26)
Fearful/Agitate	3 (0–25)	6.2 (0–19.4)	**7.7** (0–17.5)
Fall/lie	**5.8** (0–17.7)	3.4 (0–26.7)	5 (0–17.5)
Stumble	**1.4** (0–4.9)	0 (0–40)	0 (0–14.3)
Run exit	**3.8** (0–16.9)	1 (0–20)	0.8 (0–15)

Highest median(s) for each category is in bold (if >0). See Appendix A for description of how each measure was assessed. *Nutritional deaths included weight loss and mineral deficiency (e.g., phosphate) deaths. *Poisoning deaths were plant poisonings and other poisonings (e.g., urea). *Reproduction related deaths included dystocia, retained placenta, and vaginal prolapse complications.

**Table 3 animals-11-00165-t003:** Groupings by farm category (1: Commercial (*n* = 17), 2: Semi-commercial (*n* = 20), 3: Communal (*n* = 18)), using the Dunn test (with Holm-Bonferroni correction for multiple comparisons)).

Farm Grouping	Measures
1 < 2 < 3	Emaciated, poor rumen fill
1 < 2, 1 < 3, 2 = 3	Thin, horns, hair loss, ticks, mortality rate
2 < 3, 1 = 2, 1 = 3	Dirtiness
1 < 3, 2 < 3, 1 = 2	Abrasion, lameness, predator deaths
2 < 3, 2 < 1, 1 = 3	Diarrhea, reproduction deaths
1 < 2, 3 < 2, 1 = 3	Deficiency (nutritional) deaths

**Table 4 animals-11-00165-t004:** Pairwise comparisons for the Fishers exact test for the frequency of categorical measures at the cow–calf production systems herds in Namibia (1—Commercial (*n* = 17), 2—Semi-commercial (*n* = 20), 3—Communal (*n* = 18)).

Ordinal Measures	Fishers Exact *p*-Value			Pairwise Outcomes
	1 vs. 2	1 vs. 3	2 vs. 3	
Water distance	<0.001	<0.001	1	1 < 2, 1 < 3, 2 = 3
Grazing distance	<0.001	<0.001	1	1 < 2, 1 < 3, 2 = 3
Dehorning	0.001	<0.001	0.468	1 < 2, 1 < 3, 2 = 3
Castration	0.005	<0.001	0.135	1 < 2, 1 < 3, 2 = 3
Ear tagging	<0.001	<0.001	0.526	1 < 2, 1 < 3, 2 = 3
Hot-iron branding	<0.001	0.466	<0.001	1 < 2, 1 = 3, 3 < 2
Mis-catch	0.094	0.019	0.526	1 = 2 = 3
Electrical prodders	0.019	0.015	1	2 < 1, 3 < 1, 2 = 3
Tail twisting	0.013	0.005	0.639	1 < 2, 1 < 3, 2 = 3
Handlers noise	0.068	0.005	0.028	1 = 2, 1 < 3, 2 = 3
Dog noise	0.270	0.019	0.027	1 = 2 = 3
Yarding/handling freq.	0.363	<0.001	<0.001	1 = 2, 1 < 3, 2 < 3
Yard/handling flow	<0.001	<0.001	0.512	1 < 2, 1 < 3, 2 = 3

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
