# Peer review of "Assessing Extensive Semi-Arid Rangeland Beef Cow–Calf Welfare in Namibia: Part 1: Comparison between Farm Production System’s Effect on the Welfare of Beef Cows"

_animals, 2021, doi:10.3390/ani11010165_

Round 1

Reviewer 1 Report

Very good to see research in this area.

Generally well written.

63: As 'semi-commercial" is defined above as communal, what are the three types?

299-308: As regional differences may be an important factor more specific detail about the regions (recent rainfall?) should be given especially as there is a prevailing drought condition.

388: branding is not the suitable term for this procedure: "hot iron line firing"?

Author Response

Thanks so much for your valuable comments.

Reviewer 2 Report

This is a good Ms. Congratulation. Anyway some comments from my side. I found same confounder which I ask you to explain or correct, like rumen filling as a short-term score and BCS as a long-term score.

Line

Comment

31

Should this not have the meaning of: „… with five herds in on semi-commercial village level.“?

97

Please explain the meaning of Total cow/heifers/herd

107

give the range of altitude above sea level

109

give the range of altitude above sea level

123

a range cannot have <

125

a range cannot have a <

141

give the range of altitude above sea level

189

define blindness: Was it tested in an ophthalmological way or in a general clinical way? In ophthalmological studies severe cataract with major loss of visibility is found in 3% of bovines. I accept your results, but they have to be explained who the animals were tested for.

197

stick to bold

208

check this explanation of the Bonferroni -Dunn correction: To be significant the p-value has to be <0.05/3.

291

Rumen fill is a very short-term scoring. It can change very quickly. Emaciated cattle and BCS are finding with a long-term influence. This has to be mentioned in the text and this difference has to be discussed.

324

The relation of lameness to BCS is directly linked to feeding, because lameness is related to the fat cushion in the claw. Therefor you have to mention that it is a feeding problem.

348

Another confounder could be the control of cows at birth in commercial large farms and the availability of specialists, i.e. veterinarians in communal farms in case of dystocia.

371

Explain FAN (Farm Assured Namibian)

401

village farm is not consistent with the 3 farm systems mentioned above. Communal = village?

475

once you write 4109-4120 and once 4109-20, check with ANIMALS layout; check all

480

does it need pp? Check all

Author Response

Thank you so much for your valuable comments and edits.
